# The Unknown Role of Periostin in Psoriatic Epidermal Hyperplasia

**DOI:** 10.3390/ijms242216295

**Published:** 2023-11-14

**Authors:** Milena Wojciechowska, Kinga Ścibior, Monika Betyna-Białek, Ewa Kostrzewska, Oliwia McFarlane

**Affiliations:** 1Department of Social and Medical Sciences, Nicolaus Copernicus University in Toruń, Ludwik Rydygier Collegium Medicum in Bydgoszcz, 85-077 Bydgoszcz, Poland; oliwia.mcfarlane@cm.umk.pl; 2Department of Dermatology and Venerology, Nicolaus Copernicus University in Toruń, Ludwik Rydygier Collegium Medicum in Bydgoszcz, 85-088 Bydgoszcz, Poland; k.scibior@cm.umk.pl; 3Center for Specialist Languages in Medicine, Nicolaus Copernicus University in Toruń, Ludwik Rydygier Collegium Medicum in Bydgoszcz, 85-088 Bydgoszcz, Poland; monika.betyna@cm.umk.pl; 4Statistical Analysis Center, Nicolaus Copernicus University in Toruń, 87-100 Toruń, Poland; e.kostrzewska@umk.pl

**Keywords:** periostin, psoriasis, hyperplasia, pathomechanism, epidermis

## Abstract

Psoriasis is an inflammatory skin disease that affects 1–2% of the general population. The pathomechanism is based on type 1 immunological reactions. Hyperplasia of the epidermis in psoriasis is a result of disrupted epidermal architecture due to increased synthesis and expression of extracellular matrix proteins. In our study, we analyzed the involvement of periostin (POSTN) in the pathogenesis of psoriasis, as one of the extracellular matrix proteins belonging to the fasciclin family. The study group consisted of 70 patients with psoriasis, while the control group comprised 30 healthy individuals. The serum concentrations of POSTN, Il-6, Il-17, Il-22, TNF-α and IFN-γ were measured in all participants. The severity of psoriasis was determined using the PASI (Psoriasis Area and Severity Index) score. The presence of POSTN in biopsy samples of 50 patients was assessed using the direct immunofluorescence method. The results were subjected to statistical analysis. The serum concentrations of POSTN, Il-6, Il-17, Il-22, TNF-α and IFN-γ in the study group are significantly higher than in the control group. Positive correlation has been demonstrated between the PASI score and the investigated cytokines, but not with POSTN. There was no statistically significant correlation between the POSTN level and the cytokines levels. POSTN deposits were localized in the epidermis in 66% of patients with psoriasis. The role of POSTN in the pathogenesis of psoriasis remains unclear. The mechanisms inducing the synthesis and expression of POSTN in psoriatic skin are not yet fully understood. Further research is needed to enhance our understanding of the mechanism underlying epidermal hyperplasia in psoriasis.

## 1. Introduction

Psoriasis is a chronic inflammatory disease that affects approximately 1–2% of the general population. It is a systemic disease but primarily affects the skin and joints. The long-lasting nature of psoriasis significantly impairs the quality of life, disrupts the emotional well-being of patients and can lead to disability. Psoriasis is one of the most troublesome dermatoses. Often recurring and visible skin changes are a source of negative emotions and reactions in patients. Frequently, individuals with psoriasis experience marginalization and are perceived as neglected people with an infectious disease. This results in distorted self-awareness and a distorted perception of their own bodies. Such experiences of stress exacerbate the symptoms of the disease, further worsening the emotional state of patients and deepening their anxiety. The reduced quality of life for psoriasis patients contributes to the coexistence of somatic complications such as obesity, depression, sexual disorders, cardiovascular disorders and susceptibility to addiction [1].

On the skin, psoriasis is characterized by the presence of raised, well-defined, erythematous plaques covered with scales. The pathomechanism of psoriasis is very complex. In the skin, there is hyperproliferation and abnormal differentiation of keratinocytes. Immunological processes involving T lymphocytes play a role in these disorders [2]. T lymphocytes stimulate excessive keratinocyte proliferation, recruit and activate neutrophils and release inflammatory mediators. The pathogenesis of psoriasis is characterized by the dominance of Th1, Th17 and Th22 lymphocytes, as well as the release of cytokines by these cells, including Il-6, Il-17, Il-22, TNF-α and IFN-γ. Interactions between cells and the cytokines released by them result in the occurrence of acanthosis, hyperkeratosis and parakeratosis in psoriatic skin [3].

Periostin (POSTN) is a protein primarily secreted by collagen-rich connective tissues, including the skin. POSTN is an extracellular matrix protein that plays a key role in building the extracellular architecture. It belongs to the fasciclin family, and its molecular weight is approximately 90 kDa [4]. Moreover, it is an extracellular matrix protein that is involved in various physiological processes, such as wound healing, embryonic development and the proliferation and differentiation of osteoblasts. This protein is considered a matricellular protein with regulatory functions in tissue remodeling and cell–matrix interactions. Through integrins present on the cell surface, such as αvβ3 and αvβ5, POSTN transduces signals within cells. This extracellular matrix protein is essential for the proper development of heart valves, periodontal ligaments and lung remodeling in newborns. Furthermore, it regulates cell–matrix interactions during tissue repair, inflammatory processes and tumor metastasis. It plays a role in modulating the microenvironment and cellular responses in these contexts [4,5]. POSTN is involved in various pathological processes, including fibrosis following myocardial infarction, pulmonary fibrosis and skin fibrosis. The expression of POSTN is induced by growth factors (Transforming Growth Factor) TGF-β1, TGF-β2, TGF-β3, BMP-2, BMP-4, Vascular Endothelial Growth Factor (VEGF), Connective Tissue Growth Factor (CTGF 2) and interleukins (IL-3, Il-4, Il-6 and Il-13) [5]. The involvement of POSTN has been demonstrated in the development and homeostasis of periodontal diseases, kidney inflammation, osteoporosis, allergic diseases and skin disorders. It has been confirmed that overexpression of POSTN in cancer promotes metastasis and cell survival by activating the Akt/PKB pathway. POSTN, through the induction of matrix metalloproteinases (MMP-9, MMP-10 and MMP-13), drives the epithelial–mesenchymal transition (EMT), resulting in matrix degradation and facilitating local tumor spread [5,6].

In healthy adult skin, POSTN is primarily localized in the basement membrane and around the hair follicles of the dermis. By acting on keratinocytes and fibroblasts under normal physiological conditions, POSTN has a protective effect against skin damage. POSTN overexpression has been shown in the pathogenesis of some skin diseases, including atopic dermatitis (AD), psoriasis and scleroderma. In systemic sclerosis, POSTN promotes and intensifies skin fibrosis. In AD, POSTN exacerbates tissue remodeling, leading to the development of hyperkeratotic skin lesions with distorted architecture [3,7,8]. AD and psoriasis are dermatoses characterized by epidermal hyperplasia, but the immune response in each of these diseases occurs differently. In the acute phase of AD, type 2 of immune reactions dominate, while in the chronic phase, type 1 reactions prevail. In the pathogenesis of psoriasis, TNF and IFN-γ activate dendritic cells through the activation of type 1 immune pathways. IL-6, IL-17 and IL-22 play a crucial role in epidermal hyperplasia in psoriasis [9].

Research on the involvement of POSTN in the pathophysiology of psoriasis is limited. The available data are insufficient to determine the exact role of POSTN in the pathogenesis of psoriasis. Therefore, the aim of this study was to investigate whether there is a difference in POSTN levels between patients with psoriasis and a healthy control group. It was also assessed whether the values of cytokines crucial for the course of psoriasis correlate with the concentration of POSTN and may be important in the induction of POSTN synthesis.

## 2. Results

The concentration of POSTN in the study group ranges from 245.4 to 1288.7 pmol/L, with a mean value of 692.8 pmol/L. In the control group, the values range from 204.6 to 587.6 pmol/L, with a mean value of 334.3 pmol/L. The concentration of POSTN in the study group is statistically significantly higher than in the control group, as determined by the Mann–Whitney test (*p*-value = 9.882 × 10^−11^ < 0.05) (Figure 1).

The concentrations and mean values of IL-6, IL-17, IL-22, TNF-α and IFN-γ in the study group and control group are demonstrated in Table 1. The concentrations of the investigated cytokines were statistically significantly higher in the study group compared to the control group (Table 1, Figure 2).

In the study group, the PASI scores ranged from 12.5 to 34.9, with a mean value of 20.3. No statistically significant correlation was found between PASI scores and POSTN concentrations in patients with psoriasis (Pearson correlation test; *p*-value 0.0887 > 0.05). However, positive correlations were observed between PASI scores and the concentrations of IL-6, IL-17, IL-22, TNF-α and IFN-γ in the study group. Higher PASI scores were associated with higher concentrations of these cytokines (Figure 3, Table 2).

The duration of psoriasis (calculated from the year of diagnosis) in the study group ranged from 0.5 to 27 years, with a mean duration of 12.6 years. A statistically significant correlation was found between the duration of the disease and the concentration of IL-22, indicating that the longer the duration of the disease, the higher the IL-22 concentration (Pearson correlation test; *p*-value 0.0445 < 0.05). However, no such relationship was observed for the other investigated markers (Table 3, Figure 4).

In the study group, no statistically significant correlation was found between POSTN concentrations and the concentrations of IL-6, IL-17, IL-22, TNF-α and IFN-γ. However, in the control group, statistically significant correlations were observed between POSTN concentrations and the concentrations of IL-17, TNF-α and IFN-γ (Spearman correlation test; *p*-value 0.0144 < 0.05; *p*-value 0.000 < 0.05; *p*-value 0.000 < 0.05). It was found that higher concentrations of IL-17, TNF-α and IFN-γ were associated with higher POSTN levels (Table 4, Figure 5).

The presence of POSTN in skin biopsies was demonstrated in 33 (66%) out of 50 patients from whom the material was obtained. POSTN deposits were identified in the epidermis. In 17 patients (34%), the presence of POSTN was not detected in the skin biopsy (Table 5, Figure 6).

## 3. Discussion

Psoriasis is a chronic, multifactorial, autoimmune inflammatory disease. The skin lesions are characterized by well-defined, erythematous papules with adherent silvery scales. The disease fluctuates between periods of remission and exacerbation, influenced by various factors such as diet, stress, environmental factors and pathogens. Inflammatory skin conditions are accompanied by hyperproliferative reactions in the epidermis, caused by hyperactivation and production of immature keratinocytes. Psoriatic skin is characterized by a thickened epidermal layer, along with inflammatory infiltrates in the dermis primarily composed of dendritic cells, neutrophils, T lymphocytes and macrophages [7,10].

Psoriasis pathogenesis involves both innate mechanisms and adaptive immune responses occurring in the skin. Immune cells release pro-inflammatory cytokines such as TNF-α, IFN-γ and IL-1β, which, in turn, facilitate the activation of myeloid dendritic cells. These cells release IL-12 and IL-23, which are key cytokines for the differentiation of naive T cells into Th1, Th22 and Th17 cells [10,11]. In patients with psoriasis, increased activity of Th17 and Th22 cells has been confirmed compared to the healthy group. For a long time, Th17 cells were considered the dominant source of IL-22. However, it has been shown that CD4+ T lymphocytes can release IL-22 alone without IL-17, leading to their classification as Th22 cells. The differentiation of Th17 cells requires IL-6, TGF-β and IL-23. Through the activation of STAT3 and subsequent promotion of transcription of Th17-specific genes by IL-6 and TGF-β, the process of differentiating naive T cells into Th17 cells can be initiated [2]. However, Th17 cells generated through differentiation stimulated by IL-6 and TGF-β have limited pathogenicity, and further maturation and development require subsequent exposure to IL-23. As a result, pathogenic Th17 cells produce key cytokines for the development of psoriasis, including IL-17 and IL-22 [2,12]. Th17 cells participate in the activation and migration of T lymphocytes and neutrophils and enhance the activation of monocytes. Other cytokines released by Th17 cells, such as IL-6, IL-21 and TNF, initiate parakeratosis and hyperkeratotic acanthosis in psoriasis. IL-6 is indeed necessary for the initial differentiation of naive CD4+ T cells into Th17 cells. It also plays a role in the acute phase response, T cell activation and keratinocyte proliferation [1]. In the study, we demonstrated that the concentrations of IL-6, IL-17, IL-22, TNF-α and IFN-γ were statistically significantly higher in the group of patients with psoriasis compared to the healthy group. Similar results have been also confirmed in other scientific studies [11,12,13]. Furthermore, we confirmed increasing concentrations of these cytokines with the severity of psoriasis, as expressed by the PASI score. TNF-α, IFN-γ, IL-22 and IL-17 derived from Th cells sustain the inflammatory process and induce keratinocyte proliferation [14].

Psoriasis is characterized by epidermal hyperplasia and disturbed keratinocyte differentiation. Compared to normal skin, psoriatic epidermis has a lower percentage of apoptotic cells. Additionally, keratinocytes derived from psoriatic plaques are more resistant to apoptosis-inducing factors compared to normal keratinocytes. Psoriatic keratinocytes are resistant to signals mediated by TNF-α, which leads to a paradoxical increase in TNF-α levels in the skin lesions as well as serum of patients with psoriasis [15]. Proteins from the Bcl-2 family play a crucial role in the dysregulation of apoptosis in psoriasis, including both pro-apoptotic proteins (Bax, Bak and Bad) and anti-apoptotic proteins (Bcl-2 and Bcl-xL). It has been observed that psoriatic skin exhibits a lower expression of Bcl-2 and higher expressions of Bcl-xL and Bax. However, there are no significant differences in Bcl-2 expression in the epidermis or Bcl-2 levels in the serum of patients with psoriasis compared to healthy individuals. Indeed, the presence of proteins from the Bcl-2 family in the epidermis may not be the sole determinant, but rather their relative ratios and the predominance of one over the others likely play a meaningful role [16]. In psoriasis, keratinocytes produce various chemokines, such as CCL20 and CXCL10, which recruit Th1/Tc1 and Th17 cells [17]. Keratinocytes also release cytokines such as IL-23, TNF-α and TSLP (Thymic stromal lymphopoietin). These mediators stimulate immune cells to produce cytokines that, in turn, intensify the inflammatory state by inducing keratinocyte proliferation [18]. IL-22 exhibits a strong hyperproliferative effect on keratinocytes. In mouse models, it has been shown that lower levels of IL-22 are associated with a milder course of psoriasis [19]. The research has demonstrated that IL-22 concentration correlates with the duration of psoriasis, with higher levels associated with longer disease duration. As mentioned earlier, a similar relationship between IL-22 and PASI scores was also confirmed. Among all the cytokines that were investigated, only IL-22 showed a statistically significant increase with higher PASI scores.

The multifaceted pathology of psoriasis is caused by genetic and immunological changes resulting from abnormal expression of various regulatory and structural proteins. In this research, the role of POSTN in the pathogenesis of psoriasis was fully examined. POSTN was first identified 15 years ago as osteoblast-specific factor 2. Currently, POSTN is classified as an extracellular matrix (ECM) protein and is expressed in various collagen-rich tissues. It plays important biological roles in the ECM. POSTN can interact with collagen, fibronectin and tenascin-C, thereby influencing the architecture of the ECM [20]. It is involved in maintaining the integrity of the extracellular matrix and influences the biomechanical properties of connective tissue. It is also essential for tissue remodeling during development and in the repair process of inflamed tissue, with its expression significantly increasing after tissue damage. Acting as a ligand for integrins, POSTN mediates cell adhesion to the ECM and influences cell proliferation and differentiation. It is primarily produced by fibroblasts, and its expression is induced by various factors, including TGF-β1, IL-4 and IL-13 [4,9].

In the pathogenesis of psoriasis, remodeling affects the activation of immune cells, modulating the inflammatory response. In the study, it was demonstrated that POSTN levels were higher in the study group compared to the control group. The results of other scientific studies also support this relationship in skin diseases, including psoriasis and AD [9,21]. However, in the study group, a correlation between POSTN levels and the concentrations of important cytokines for psoriasis (e.g., IL-6, IL-17, IL-22, TNF-α and IFN-γ) were not confirmed. Interestingly, such correlations were observed in the control group, confirming associations between POSTN and IL-17, TNF-α and IFN-γ concentrations. IL-17 is critical for epidermal hyperplasia in psoriasis, so it is surprising that the correlation between POSTN and IL-17 was confirmed in the control group but not in the study group. It appears that POSTN does not play a significant role in the production of pro-inflammatory cytokines, and the inflammatory mediators crucial for psoriasis do not influence POSTN synthesis. Similar findings were obtained by Arima et al. [9] in their experiment using a mouse model with topical treatment of imiquimod (IMQ). This research team demonstrated that IL-17, IL-21 and IL-22 do not induce POSTN expression in fibroblasts. In their immunohistochemical analysis, Arima et al. showed the presence of POSTN in the dermis, mainly at the border with the epidermis. This result differs from the findings of this research, as in over half of the skin biopsies we assessed, POSTN was localized in the epidermis, and in some patients, POSTN presence in the biopsies was not detected at all. Similar observations were made by Flink et al. [22], confirming the increased expression of POSTN in the basal layer of the epidermis, as well as in the serum of psoriasis patients. This group of researchers concluded that keratinocytes in the basal layer of the epidermis play a key role in the enhanced production of POSTN in psoriatic skin. Given the discrepancy in the results of the few studies on this topic, further research is needed to better assess the role of POSTN in the pathophysiology of psoriasis. Further research is needed to determine the mechanism of POSTN migration to the upper layers of the skin. The absence of POSTN in the biopsies may be due to the fact that a biopsy taken from a single, largest skin lesion was evaluated. It is possible that the presence of POSTN would be demonstrated in the material obtained from a different skin lesion in the 17 POSTN-negative patients. Perhaps this methodology in our study limited the ability to obtain a comprehensive answer to all questions regarding the role of POSTN in the pathophysiology of psoriasis. No statistical significance between POSTN levels and the PASI scores was also found, indicating a lack of correlation between POSTN synthesis and the severity of psoriasis. This result further underscores the unclear role of POSTN in the pathogenesis of psoriasis.

## 4. Materials and Methods

The study was conducted in full compliance with all relevant institutional and governmental regulations regarding the ethical involvement of human volunteers as well as in accordance with the principles outlined in the Helsinki Declaration. The study protocol was approved by the institutional review board, specifically the Bioethics Committee of the University of Nicolaus Copernicus in Toruń, Ludwik Rydygier Collegium Medicum in Bydgoszcz, Poland (approval no. 200/2019). All participants included in the study provided written informed consent prior to their involvement.

### 4.1. Study Subjects

The study group consisted of 70 patients with psoriasis hospitalized in the Clinic of Dermatology and Venereology of the Antoni Jurasz Clinical Hospital of the Nicolaus Copernicus University in Toruń, Ludwik Rydygier Collegium Medicum in Bydgoszcz, Poland. There were 33 women and 37 men aged 22–69 years (mean age 44.5 years) in the study group. The control group consisted of 30 healthy participants, including 17 women and 13 men, aged 20–55 years (mean age 40.8 years).

The study group was selected from hospitalized patients with psoriasis. The sampling followed strict inclusion criteria: (1) clinical diagnosis of psoriasis confirmed by a dermatologist, (2) the lack of systemic treatment for psoriasis for a minimum of 3 months prior to the study (topical or systemic, anti-inflammatory drugs, immunosuppressive drugs, inhibitors that inhibit the activity of targeted cytokines) and (3) age 18 or older. Exclusion criteria were as follows: (1) refusal to participate in the study and (2) active dermatological diseases other than psoriasis.

### 4.2. Laboratory

Venous blood was collected from both groups according to standard laboratory procedures. The serum concentrations of POSTN, IL-6, IL-17, IL-22, TNF-α and IFN-γ were determined using the ELISA method (Biorbyt Ltd., Cambridge, UK). Blood samples were collected on the first day of hospitalization before the patients received any necessary treatment that could potentially interfere with the test results.

### 4.3. Skin Biopsies

Skin biopsies were performed on 50 patients with psoriasis. The material was obtained from a single, largest psoriatic lesion using a 5 mm diameter punch biopsy tool. To identify POSTN in the examined tissue, the direct immunofluorescence method was implemented. In the first stage of the study, primary Anti-Periostin antibodies (Biorbyt Ltd., Cambridge, UK) were utilized, followed by secondary IgG antibodies conjugated with fluorescein isothiocyanate (FITC) (Biorbyt Ltd., Cambridge, UK). Skin biopsy was performed on 50 out of 70 participants, as 20 patients did not consent to this procedure. This does not significantly impact the results of our study since we were able to draw meaningful conclusions by evaluating the material obtained from the 50 participants. Skin biopsy was not conducted in the control group because the Bioethical Commission did not grant permission for the collection of this material from healthy individuals. This might be a limitation of the study, as comparing the material between the experimental and control groups could have provided valuable data.

### 4.4. Dermatological Assessment

The Psoriasis Area and Severity Index (PASI) was used to assess the severity of psoriasis. The maximum PASI value was 72, and the minimum was 0. Higher scores indicated greater severity of the clinical lesions.

### 4.5. Statistical Analyses

In order to conduct the analyses, R programming language version 4.3.0 and the following packages were used: cowplot (1.1.1), ggpubr (0.6.0), gridExtra (2.3), ggplot2 (3.4.2), dplyr (1.1.2), stringr (1.5.0) and readxl (1.4.2). To compare continuous variables in two groups, two statistical tests were utilized: the two-sided Mann–Whitney test and the two-sided unpaired Student’s t-test. To examine the relationships between continuous variables, two statistical tests were used: the two-sided Pearson test and two-sided Spearman test. A significance level of 0.05 was adopted.

## 5. Conclusions

The involvement of POSTN in epidermal hyperplasia in psoriasis appears to be undeniable. However, further research is needed to elucidate the mechanisms underlying the increased synthesis of POSTN in psoriasis. The lack of involvement of key cytokines in relation to POSTN activity suggests the involvement of other factors or their combinations. Conducting further studies may lead to the discovery of new mechanisms regarding the role of POSTN in the pathogenesis of psoriasis.

## Figures and Tables

**Figure 1 ijms-24-16295-f001:**
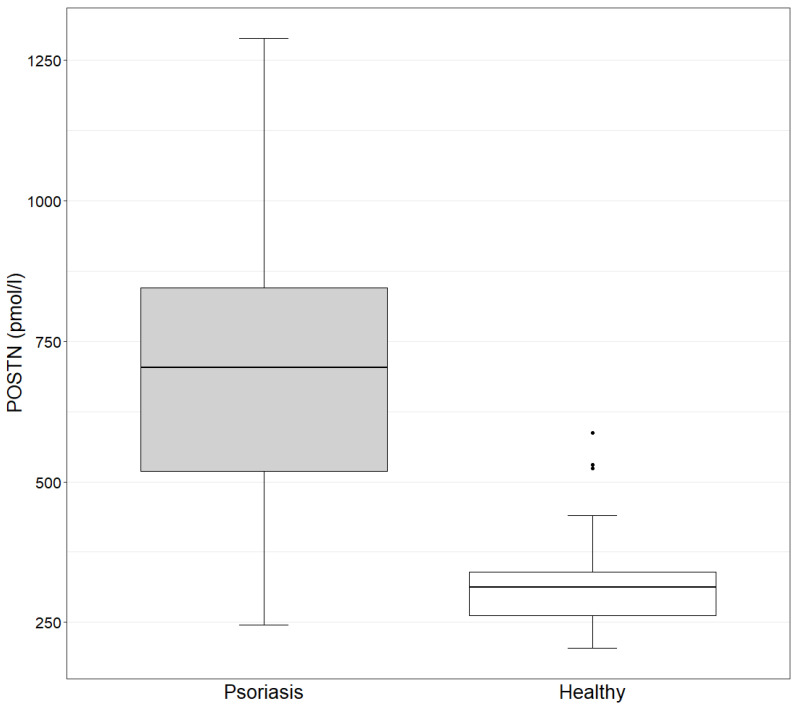
POSTN concentration in the study and control groups (Mann–Whitney test). Study group: POSTN mean = 692.8 (245.4–1288.7); median = 702.7. Control group: POSTN mean = 334.3 (204.6–587.6); median = 312.

**Figure 2 ijms-24-16295-f002:**
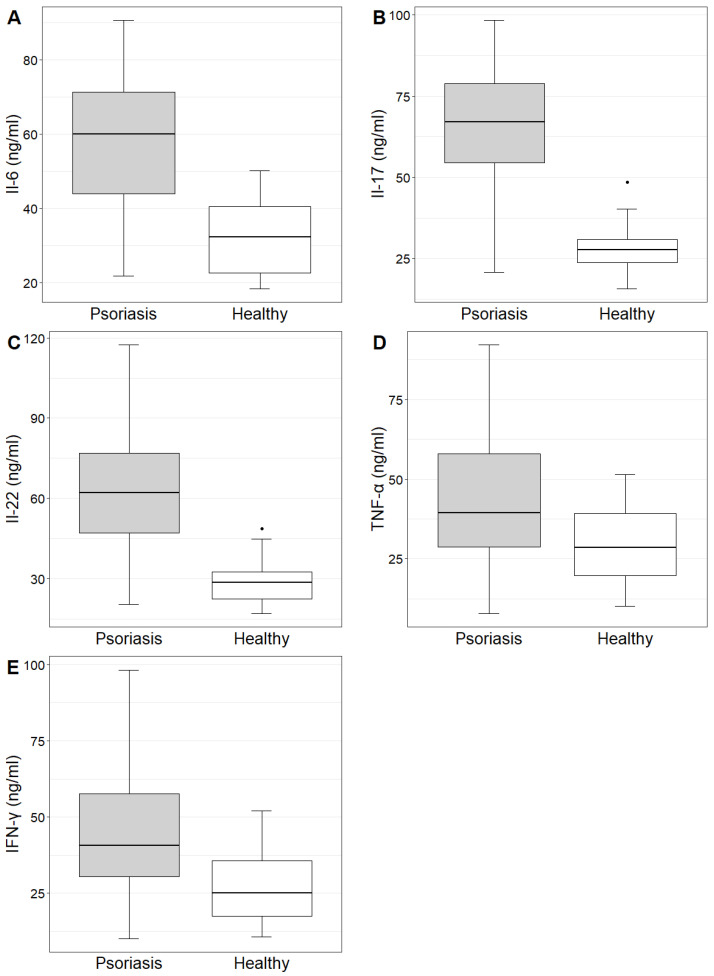
Il-6, Il-17, Il-22, TNF-α and IFN-γ (figures (**A**–**E**), respectively) concentrations in the study and control groups.

**Figure 3 ijms-24-16295-f003:**
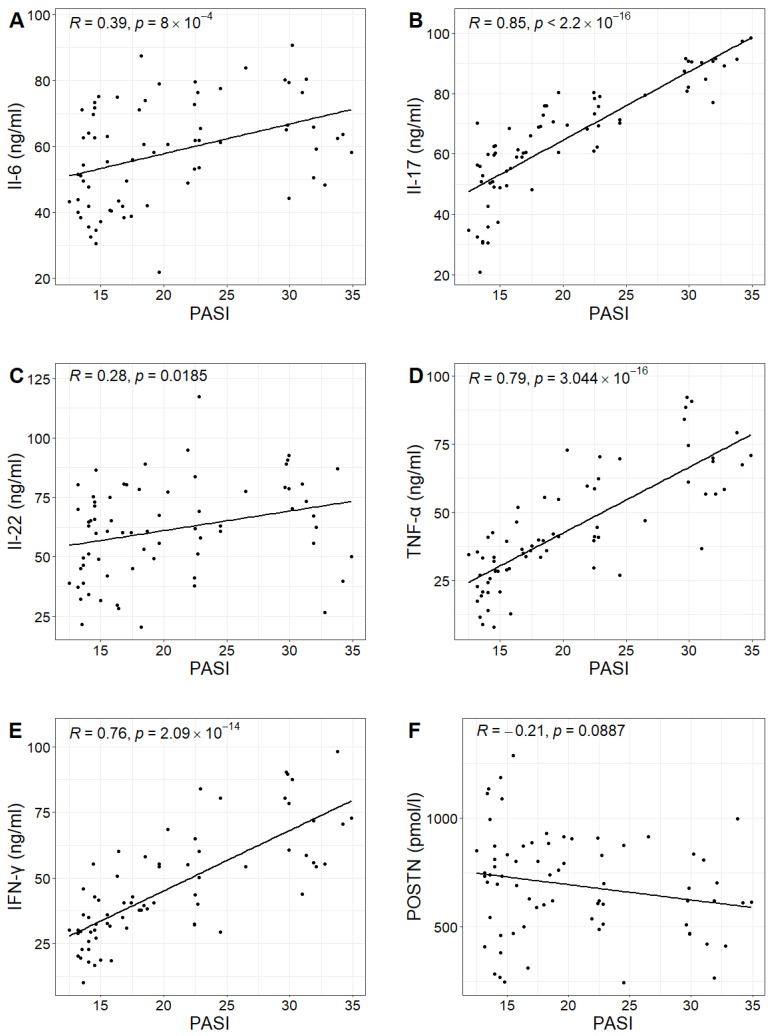
Relationship between the concentrations of Il-6, Il-17, Il-22, TNF-α, IFN-γ and POSTN (figures (**A**–**F**), respectively) and the PASI values in the study group with Pearson correlation coefficient R and *p*-value of Pearson correlation test.

**Figure 4 ijms-24-16295-f004:**
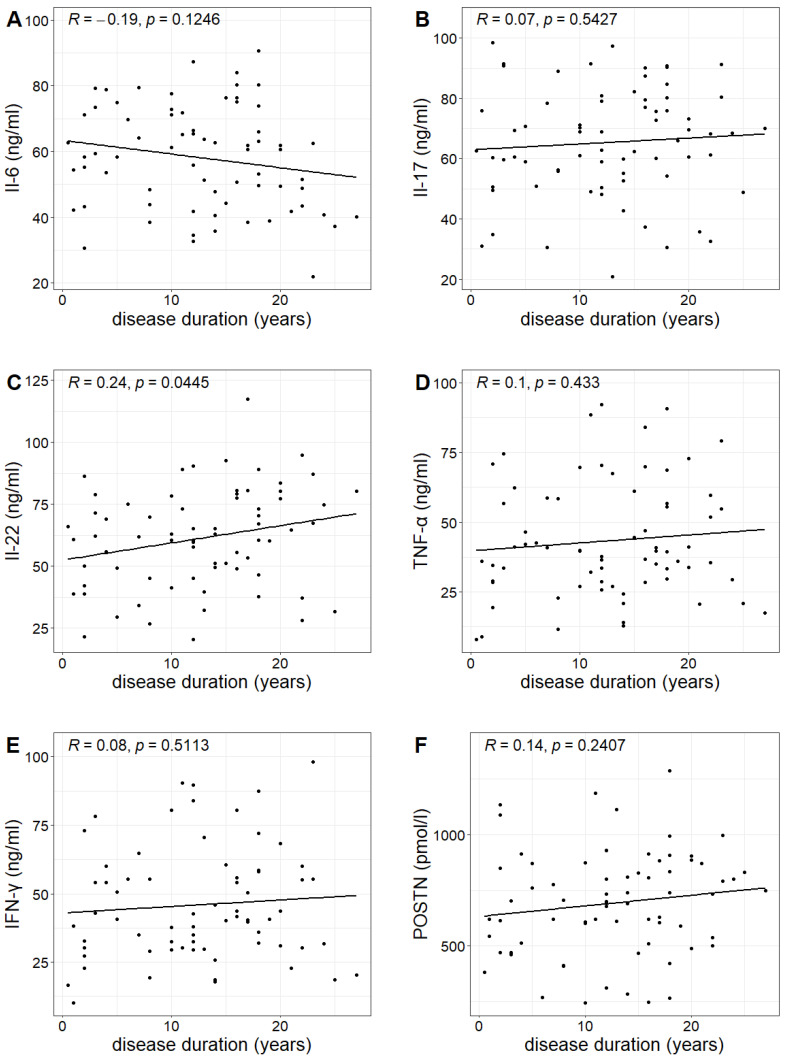
Relationship between the concentrations of Il-6, Il-17, Il-22, TNF-α, IFN-γ and POSTN (figures (**A**–**F**), respectively) and the duration of psoriasis with Pearson correlation coefficient R and *p*-value of Pearson correlation test.

**Figure 5 ijms-24-16295-f005:**
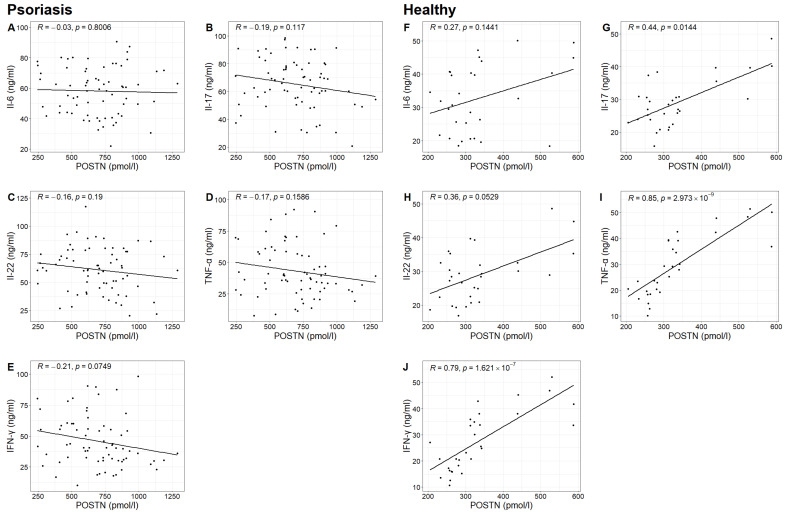
Relationship between the concentrations of Il-6, Il-17, Il-22, TNF-α and IFN-γ and the concentration of POSTN in the study group (figures (**A**–**E**), respectively) with Pearson correlation coefficient R and *p*-value of Pearson correlation test and in the control group (figures (**F**–**J**), respectively) with Spearman correlation coefficient R and *p*-value of Spearman correlation test.

**Figure 6 ijms-24-16295-f006:**
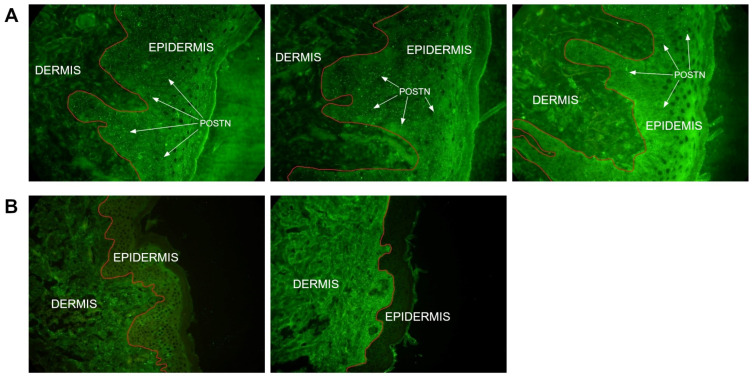
Analysis of POSTN localization in skin biopsy: (**A**) POSTN in the epidermis, and (**B**) absence of POSTN in the skin biopsy. In part A, POST is visible in the form of white immunofluorescence markers. The microscope’s magnification and scale bar are the same. Microscope scale 400×.

**Table 1 ijms-24-16295-t001:** Concentrations and mean values of Il-6, Il-17, Il-22, TNF-α and IFN-γ (ng/mL) in the study and control groups (Student’s *t*-test).

ng/mL	Group	Minimum	Median	Mean	Maximum	*p*-Value
Il-6	study	21.9	60	58.1	90.6	4.76 × 10^−15^
	control	18.4	32.3	32.7	50.1	
Il-17	study	20.8	67.1	65.3	98.4	1.607759 × 10^−25^
	control	15.7	27.7	29.0	48.6	
Il-22	study	20.3	62.1	61.3	117.4	4.203961 × 10^−20^
	control	16.9	28.7	28.9	48.6	
TNF-α	study	7.8	39.5	43.3	92.1	8.594 × 10^−5^
	control	10.2	28.6	29.7	51.4	
IFN-γ	study	10.2	40.6	45.9	98.2	1.534 × 10^−7^
	control	10.7	25.2	27.5	52.1	

**Table 2 ijms-24-16295-t002:** Pearson correlation test results between concentrations of investigated markers and PASI values in the study group.

	POSTN	Il-6	Il-17	Il-22	TNF-α	IFN-γ
correlation coefficient	−0.2050	0.3908	0.8507	0.2808	0.7924	0.7610
*p*-value	0.0887	0.0008	0.0000	0.0185	0.0000	0.0000

**Table 3 ijms-24-16295-t003:** Pearson correlation test results between concentrations of investigated markers and duration of psoriasis.

	POSTN	Il-6	Il-17	Il-22	TNF-α	IFN-γ
correlation coefficient	0.1421	−0.1853	0.0740	0.2410	0.0952	0.0798
*p*-value	0.2407	0.1246	0.5427	0.0445	0.4330	0.5113

**Table 4 ijms-24-16295-t004:** The result of the correlation test between the concentrations of the tested cytokines and the concentrations of POSTN (Pearson correlation test; Spearman correlation test).

		Il-6	Il-17	Il-22	TNF-α	IFN-γ
Study group	Pearson correlation coefficient	−0.0307	−0.1891	−0.1585	−0.1704	−0.2142
	*p*-value	0.8006	0.1170	0.1900	0.1586	0.0749
Control group	Spearman correlation coefficient	0.2732	0.4424	0.3568	0.8493	0.7942
	*p*-value	0.1441	0.0144	0.0529	0.0000	0.0000

**Table 5 ijms-24-16295-t005:** Presence of POSTN in skin biopsy.

	Psoriasis (N = 50)	
	N	%
POSTN^+^	33	66
POSTN^−^	17	34

## Data Availability

Data are contained within the article.

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
