# Peer review of "The Unknown Role of Periostin in Psoriatic Epidermal Hyperplasia"

_ijms, 2023, doi:10.3390/ijms242216295_

Round 1
Reviewer 1 Report
Comments and Suggestions for Authors
1. First of all, the manuscript should be corrected for spelling and syntax errors. For example in lines 50-85 the word POSTN appears 12 times and the word periostin 6 times, which is tiresome for the reader. By the way, since you abbreviate periostin as postn, you should use postn in the text, not both of them. In line 110: just use treatment since you then define it as both systemic or topical. In line 111: either use "age above 18" or "age 18 or older" etc. etc.
2. The Methods should be presented more clearly and detailed. When and how the data were collected? Retrospectively? Prospectively? Why you performed 50 and not 70 biopsies?Wouldn't it be better if you had biopsies from the control group also? etc.
3. The readers want to know the limitations of the study to have a better understanding of it. You mention no limitations at all.
4. What are the white arrows show in photo 1A? It is not clear where POSTN is. Is it those small black circles? Explain it better please.
5. Do you have any ideas that explain the results, such as the localization of POSTN in biopsies, the relation between POSTN and interleukines in the control group etc?
6. Line 270 is an exaggeration. If its role was fully examined we would not have any unanswered questions about its role in psoriasis.
Comments on the Quality of English Language-
Reviewer 2 Report
Comments and Suggestions for Authors
The article titled ‘The unknown role of periostin in psoriatic epidermal hyper-plasia’ may be an useful contribution to the journal; however, few changes should be taken into consideration:
Periostin should be introduced in Abstract background section, connecting it to the extracellular matrix proteins, in the benefit of the reader.
In the study participants section, it is unclear whether the included patients or controls were following systemic treatments for other diseases, including immunosupressants or anti-inflammatory medication, or not; this is important and should be clarified in the manuscript, accordingly.
Also, if the psoriatic patients using topical treatment, this should be stated, and also the risk of topical (sometimes extensive) treatment to interfere with further biological tests used in the study. Authors should comment upon this and should cover this aspect, as well.
Grammar and punctuation must also be carefully checked within the entire article.
Comments on the Quality of English LanguageGrammar and punctuation must also be carefully checked within the entire article. Minor editing needed.
Round 2
Reviewer 1 Report
Comments and Suggestions for Authors
Just delete either line 295 or 299 because they are the same